# Reformulating Couscous with Sprouted Buckwheat: Physico-Chemical Properties and Sensory Characteristics Assessed by E-Senses

**DOI:** 10.3390/foods12193578

**Published:** 2023-09-26

**Authors:** Gabriella Giovanelli, Andrea Bresciani, Simona Benedetti, Giulia Chiodaroli, Simona Ratti, Susanna Buratti, Alessandra Marti

**Affiliations:** Department of Food, Environmental and Nutritional Sciences (DeFENS), Università degli Studi di Milano via G. Celoria 2, 20133 Milan, Italy; gabriella.giovanelli@unimi.it (G.G.); andrea.bresciani@unimi.it (A.B.); simona.benedetti@unimi.it (S.B.); giulia.chiodaroli@unimi.it (G.C.); simona.ratti@unimi.it (S.R.); alessandra.marti@unimi.it (A.M.)

**Keywords:** germination, buckwheat, couscous, sensory traits, phenolic profile, physical properties

## Abstract

In the frame of reformulating food products for valorizing underutilized crops and enhancing both the nutritional and sensory characteristics of traditional foods, this study explored the potential impact of sprouting on some features of couscous prepared from buckwheat. Specifically, the impact of two sprouting times (48 h and 72 h) and two enrichment levels (25% and 50%) on physical properties (bulk density, hydration properties), cooking behavior (e.g., texture), chemical features (e.g., total phenolic content, rutin and quercetin), antioxidant activity (DPPH assay), and sensory traits (by means of electronic nose, tongue, and eye) was considered. Results showed that the replacement of 50% of pre-gelatinized buckwheat flour with 72 h-sprouted buckwheat flour resulted in a couscous with a higher content of phenolic compounds (including rutin and quercetin) and antioxidant activity; the related values further increased upon cooking. Moreover, except for the hardness and gumminess that were worsened (i.e., their values increased), cohesiveness and resilience improved in the presence of sprouted buckwheat (i.e., their values increased). Finally, the overall sensory traits improved with the addition of 50% sprouted buckwheat, since both bitterness and astringency decreased in the reformulated couscous.

## 1. Introduction

Sprouting is a traditional process widely used at household level on pulses to reduce their antinutrient compound content and provide a pleasant sweet note [1,2]. The positive effect of sprouting on various compositional and nutritional traits, including the increase in minerals, vitamins, and compounds with antioxidant activity, has also been shown in cereals and pseudocereals [3,4]. The molecular changes that affect the nutritional traits of sprouted grains are triggered by the synthesis of enzymes during the sprouting process. At the same time, a controlled synthesis of amylases and proteases has been shown to be effective in increasing bread volume and reducing bread staling [5]. Thus, the study of sprouting under controlled conditions has drawn the interest of the food industry to produce new ingredients with enhanced nutritional and sensory traits to be used in baked goods.

Although the positive effects of sprouting on compositional and nutritional traits have been widely assessed, besides the increase in sweetness thanks to sugar production, very little is known about the sensory features of the sprouted grains. A recent study pointed out that sprouting was effective in decreasing the bitterness and astringency of quinoa seeds as a result of the decrease in saponin content [6]. Such chemical and sensory changes might favor the use of quinoa as seeds or as flour in wheat bread formulations [7].

Bitterness and astringency are also the sensory descriptors of buckwheat products [8] that negatively influence consumer preferences [9]. Rutin hydrolysis into quercetin by means of rutinosidase activity has been identified as one of the main mechanisms for the generation of strong bitterness in Tartary buckwheat [10]. In addition, rutin and quercetin, other phenolics are present in buckwheat seeds [11,12] and likely contribute to the sensory traits of buckwheat [13]. Among the pre-treatments proposed to enhance the sensory traits of buckwheat, in a previous study we showed that pre-gelatinization treatment was effective in decreasing the bitterness and astringency of both tartary and common buckwheat flour [14]. Since germination produces dramatic changes in the phenolic profile of buckwheat seeds, enhancing the antioxidant potential of the derived products [15,16,17], modifications in the sensory traits of sprouted buckwheat are likely expected and are explored in the present study.

Taking into consideration the several health benefits attributed to buckwheat (e.g., plasma cholesterol level reduction, neuroprotection, anticancer, anti-inflammatory, antidiabetic effects, and improvement of hypertension conditions [18]), the incorporation of buckwheat in food formulations has increased in the last few years. Thus, more and more functional foods have been developed and their quality improved, including gluten-free bread, cookies, pasta, noodles [19], and, more recently, an Italian traditional product called “polenta” [8]. In the present study, we explore—as a possible way to promote the use of buckwheat—its utilization in couscous, a traditional dish widespread in Western countries that requires low water and energy input for its cooking.

In this context, the present study aimed to investigate the effect of sprouting on the physico-chemical, technological, and sensory features of a 100% buckwheat couscous formulation. Two germination times (48 h and 72 h) and two enrichment levels (25% and 50%) were considered with the aim of improving the sensory traits of the product evaluated by e-senses (e-tongue, e-nose, and e-eye), which are powerful tools to obtain objective measurements in a short time [20,21]; moreover, the relations between sensory characteristics and phenolic profile (total phenolic, rutin, and quercetin content) were investigated.

## 2. Materials and Methods

### 2.1. Samples

Couscous samples were produced by Molino Filippini S.r.l. (Teglio, Sondrio, Italy). In addition, the control sample (CTRL) made from 100% pre-gelatinized buckwheat flour, four new formulations were tested. Specifically, 25% or 50% of pre-gelatinized flour was substituted with flour from buckwheat that was sprouted for 48 h and 72 h (Table 1).

Sprouting was carried out in a climate chamber (IPP110ecoplus, Memmert GmbH Co. KG, Schwabach, Germany). Dehulled common buckwheat (Fagopyrum esculentum) (2.5 kg) was soaked in water (seeds:water ratio of 1:3 *w*/*w*) for 16 h at 27 °C, and, after removing excess water, seeds were sprouted at 27 °C and 90% relative humidity for 48 h and 72 h. After that, sprouted seeds were dried at 50 °C for 8 h in an oven (Self Cooking Center^®^, Rational International AG, Landsberg am Lech, Germany). After milling (that was carried out in a M20 Universal Mill, IKA, Werke Staufen, Germany), the fine fraction (<0.5 mm) was mixed to a commercial pre-gelatinized buckwheat flour and used for couscous production that was carried out on a pilot plant set up by Molino Filippini S.r.l. (Teglio, Sondrio, Italy). Specifically, buckwheat (pregelatinized flour or its blends with sprouted samples) was mixed with water at a 40% hydration rate. The mixture was first extruded and then steam-cooked for 20 min at 80 °C and dried for 20 min at 85 °C to a final moisture content of 12%.

Couscous was used raw (R) or cooked (C), as specified below. Cooking was carried out by adding boiling water to the couscous in a 1:1 (*w*/*w*) ratio and waiting for 5 min, during which the water was completely absorbed. For phenolic profile and antioxidant activity (Section 2.4) and e-sense analysis (Section 2.5), cooked couscous was frozen at −20 °C, freeze-dried (−80 °C for 72 h), and ground using a mortar and pestle.

### 2.2. Physical Properties

The bulk density of raw couscous was measured according to Debbouz and Donnelly [22]. The volume of 50 g of couscous was determined in a 100 mL graduated cylinder. The bulk density was expressed as the ratio between weight and volume (g/mL).

The Water Absorption Index (WAI) and Water Solubility Index (WSI) were assessed both on raw and cooked couscous following the method described by Anderson et al. [23] with some modifications. In a pre-weighted 50-mL Falcon tube, 2.5 g of couscous was suspended in 30 mL of distilled water at 30 °C. After 30 min of mixing, the samples were centrifuged with a Rotofix 32A (Andreas Hettich GmbH & Co. KG, Tuttlingen, Germany) at 2500× *g* for 12 min. The supernatant was poured into pre-weighted Petri dishes and incubated in the air oven at 115 °C for 24 h. The sediment was weighed immediately, while the dry residue in the Petri dishes was measured after 16 h of drying. WAI was calculated by subtracting the weight of the residual solid after drying from the ratio between the weight after hydration and the initial dry weight. WSI is expressed in percentage and represents the ratio between the weight of the dry residual solid and the initial sample weight.

Weight Increase (WI) and Volume Increase (VI) were measured by weighing 10 g of raw couscous in a 50 mL graduated cylinder. After noting the initial volume, the couscous was cooked for 5 min, adding 10 mL of boiling water and covering the cylinder with aluminum foil. The weight and volume of the cooked samples were measured, and the differences were calculated and expressed as a percentage.

All the measurements were carried out in triplicate.

### 2.3. Textural Properties

The texture of cooked couscous was measured according to Aboubacar & Hamaker [24], with some modifications. Texture Profile Analysis (TPA) was assessed using a TA-XT plus texture analyzer (Stable Micro Systems Ltd., Godalming, UK) equipped with a 100 N load cell and a P/30 compression probe. Approximately 5 g of cooked couscous in a round plastic container (10 mm height and 40 mm diameter) were compressed until 50% of the original height at a speed of 1 mm/s. Two compression cycles occurred at a distance of 5 s. The parameters considered were Hardness (i.e., maximum force at the first compression, expressed in N), Cohesiveness (i.e., the ratio between the area of the second compression and the area of the first compression), Gumminess (i.e., hardness × cohesiveness), and Resilience (i.e., the ratio between the area after the peak of the first compression and the area before the peak of the first compression). For each sample, three independent cooking trials were carried out, and seven subsamples from each cooking trial were analyzed.

### 2.4. Phenolic Profile and Antioxidant Activity

Free phenolics were extracted from both raw and cooked samples as previously described [14]. Total phenolic content was quantified using the Folin-Ciocalteau assay; rutin, quercetin, and other flavonoids were determined by HPLC; and total antioxidant activity was determined by the DPPH assay following the methods already reported [14]. For each sample, extractions were carried out in triplicate; results were expressed on a dry weight basis, determining the moisture content by the gravimetric method [25].

### 2.5. E-Senses

#### 2.5.1. E-Nose

The volatile profile of cooked couscous was assessed by the commercial e-nose (PEN3; Win Muster Airsense Analytics Inc., Schwerin, Germany) composed of ten MOS sensors [14]. For the analysis, 0.5 g of samples were placed in 40 mL Pyrex^®^ vials fitted with a pierceable silicon/Teflon disk in the cap. For the development of the headspace, samples were kept for 2 h at room temperature and for 10 min at 35 °C ± 1 °C. During measurement, the headspace was pumped over the sensor surfaces for 60 s (injection time) at a flow rate of 400 mL/min and the sensor signals were taken at 50 s. After each determination, sensors were purged for 600 s with filtered air, and the sensor baselines were re-established for 5 s. Each sample was evaluated in duplicate, and the average values were collected and submitted for statistical elaboration.

#### 2.5.2. E-Tongue

Analyses were performed on cooked couscous samples with the Taste-Sensing System SA 402B (Intelligent Sensor Technology Co., Ltd., Atsugi, Japan) [26]. The detecting sensors used in this work were CT0 for saltiness, C00 and CA0 for bitterness and aftertaste bitterness, AE1 for astringency and aftertaste astringency, and AAE for umami. Three grams of the cooked couscous samples were added to 120 mL of distilled water, stirred by Ultraturrax for 2 min at 15,000 rpm, and centrifuged (2800× *g*) for 5 min at room temperature. The supernatant was filtered and then analyzed by e-tongue, applying the procedure reported by Laureati et al. [20]. Each sample was evaluated in duplicate, and the sensor outputs were converted to “taste values” by using appropriate coefficients based on the Weber-Fechner law, as reported by Kobayashi et al. [26].

#### 2.5.3. E-Eye (Colorimeter)

The color of cooked couscous samples was measured using a tristimulus colorimeter (CR 210, Minolta Co., Osaka, Japan). Results were expressed in the CIE L*a*b* color space, and the following indices were considered: 100-L* as the browning index and the a*/b* ratio as a measurement of the yellowness and redness. The head of the colorimeter was directly put on the surface of couscous samples and placed in a tray, and the measurements were taken at five points in order to obtain the average value for L*, a*, and b*.

### 2.6. Statistics

One-way analysis of variance (one-way ANOVA) and Multiple Range Test (Fisher’s LSD, at 95% significance level) were applied to analytical data to evidence statistically significant differences between the samples using Statistica Centurion v. 18 software (Statistical Graphics Corp., Herndon, VA, USA). Moreover, data were transformed by column autoscaling and explored by Principal Component Analysis (PCA) using the XLSTAT v. 3.1 (Addinsoft 2021, New York, NY, USA) software.

## 3. Results

### 3.1. Physical Properties of Raw Couscous

Experimental couscous showed a high variability in bulk density, ranging from 0.623 to 0.706 g/mL (Table 2). These results are comparable to the results reported for durum wheat couscous [22]. Overall, the replacement of buckwheat flour with sprouted buckwheat resulted in a product with a higher bulk density only at the lowest enrichment level (i.e., 25%), regardless of the sprouting time.

The bulk density of couscous is influenced by both the granule compactness and the vacant spaces between the granules. The former is affected by both the raw material composition and the processing technology; the latter is affected by the granule size and shape, which affect the amount of air entrapped between the granules [27]. Moreover, a relationship between bulk density and particle size was shown [28]. Samples at the 25% enrichment level indeed exhibited the highest percentage of particles greater than 1000 μm (Appendix A). In the case of samples containing sprouted buckwheat, the bulk density increased according to the sprouting time. Differences in bulk density among the samples might impact sample handling: an increase in the index suggests that the same weight of couscous would need more storage and transportation space, which might affect the handling costs [28]. Moreover, the bulk density might affect the cooking properties; usually, high granule compactness requires longer rehydration times for the product [22,27]. In this study, hydration properties were evaluated before and after cooking. Results on raw samples are reported in Table 2.

Water absorption index (WAI) and water solubility index (WSI) provide information about the amount of water that a sample can absorb and the amount of sample material that solubilizes in water, respectively. WAI is considered a positive attribute for raw couscous since it is an indicator of the ability to absorb water during cooking [22]. On the contrary, WSI is a negative attribute since it indicates the extent to which couscous disintegrates during cooking [27]. A decrease in WAI and an increase in WSI were observed in samples containing sprouted buckwheat (Table 2). While the WAI of the blends with sprouted buckwheat was lower than the values commonly found in the literature [22,24], the WSI values were comparable, as they normally range between 4 and 16% [22]. Changes in hydration properties of sprouted buckwheat-enriched couscous could be explained by the degradation of large biopolymers during sprouting, which results in a lower ability to bind water and a higher amount of short-soluble molecules [29]. Indeed, in the case of buckwheat flour, a decreased ability to bind water was observed and related to the degradation of starch and proteins [30].

### 3.2. Physical and Textural Properties of Cooked Couscous

Among the functional properties, the weight and volume changes during cooking provide an overview of the couscous cooking behavior. Specifically, weight increase (WI) and volume increase (VI) are indicators of the absorption of water and swelling of the granules. WI was not affected by sprouting (Table 3), suggesting that all the samples were able to absorb water for around 90% of their weight. On the other hand, the volume of the samples significantly increased when sprouted buckwheat was included in the formulation, regardless of the level of enrichment or the sprouting time. As regards the hydration properties, WAI significantly decreased in the presence of sprouted buckwheat, reaching the lowest value in the case of S72h_25-C. Finally, sprouted buckwheat enrichment resulted in an increase in WSI, suggesting the formation of a structure characterized by a low ability to hold the component. Overall, sprouting time did not significantly affect the functional properties of couscous; on the other hand, the enrichment level affected both the WAI (but only for prolonged sprouting time) and WSI (when sprouting was carried out for 48 h).

Data on the textural properties of cooked couscous are reported in Table 4. Good-quality couscous is a soft, non-sticky granular product with pleasant sensory attributes [27]. It has been observed that the use of different grains or different varieties can influence the texture properties of couscous [24,31].

Hardness represents the peak force at the first compression and can be compared to the force applied at the first bite. Couscous is expected to show low hardness values, as good-quality couscous is supposed to be soft [27]. Couscous containing sprouted buckwheat generally resulted in higher hardness, except for sample S48h_50-C. The hardness of couscous is linked to the granulometry: the finer the granules, the lower the hardness [32]. This might be a reason why S48_50-C showed the lowest hardness; from particle size analysis, this sample resulted in the finest (Appendix A). Cohesiveness is the ability of a product to withstand compressive or tensile stress. The samples containing sprouted buckwheat showed higher values than CTRL-C. Gumminess is calculated by multiplying hardness by cohesiveness. The unsprouted sample showed significantly lower gumminess compared to all the sprouted samples. Resilience indicates how well the product regains its original height after compression. The addition of germinated buckwheat increased the resilience compared to the CTRL-C. Semolina couscous usually shows lower values of resilience (0.17–0.23) depending on the production technology, and the substitution with different grains decreases this value [31,33].

### 3.3. Phenolic Profile Andantioxidant Activity

Total phenolic content (TPC), individual flavonoids, and total antioxidant activity of raw and cooked couscous are summarized in Table 5.

Concerning total phenolic content (TPC), two trends were observed: (i) the concentrations increased with both the sprouting time and the percentage of sprouted buckwheat added to the couscous formulation; (ii) cooked couscous showed a higher TPC value than the corresponding raw couscous (Table 5). Among the raw samples, the lowest and highest TPC were detected in CTRL-R and S72h_50_R (62 ± 9 and 256 ± 14 mg/100 g dw, respectively), corresponding to a 4-fold increase. Amongst the cooked couscous, the lowest TPC was found in CTRL-C (138 ± 6 mg/100 g dw), which was almost 2-fold higher than CTRL-R; the total phenolics content in the cooked couscous containing sprouted buckwheat ranged between 331 and 414 mg/100 g dw (about 1.5 to 4.4 times higher than the related raw samples).

The antioxidant activity of the samples, determined as free radical scavenging capacity by the DPPH assay, ranged from 148 ± 25 to 1148 ± 30 µmol TE/100 g dw and was strongly correlated to the phenolic content (linear correlation r = 0.97, Appendix A), confirming previous findings [14,15,16,34]. Consequently, couscous enriched in sprouted buckwheat flour showed a higher antioxidant activity than the CTRL-R and was dependent on both the sprouting time and the enrichment level; likewise, cooked couscous showed higher antioxidant activity than the raw samples. The increase in total phenolic content and free radical scavenging capacity as a consequence of sprouting in buckwheat has already been reported [15,34]. Flavonoids, especially rutin, represent the major phenolic components in buckwheat [35]. A significant increase in total flavonoids, including rutin and quercetin, during the germination of buckwheat seeds has been reported [15,16]. The HPLC analysis of the methanolic extracts allowed the identification and quantification of rutin and quercetin (Table 5), whose concentrations in the samples were affected by the presence of sprouted buckwheat. Thus, the amount of rutin increased from about 2 mg/100 g dw (CTRL-R) to almost 13 mg/100 g dw in S72h_50-R; quercetin showed much lower concentrations, ranging from not detectable in CRTL-R to approximately 1.2 mg/100 g dw in S72h_50-R. In accordance with literature data, new peaks referable to flavonoids appeared in the HPLC profile of couscous enriched with sprouted buckwheat: these components are referred to as “other flavonoids” (Table 5) and can be tentatively identified as 4-C-glycosylflavones vitexin, isovitexin, orientin, and isoorientin, according to their elution time and maximum absorbance [15,36]. These compounds were not detectable in the CTRL couscous and increased dramatically during sprouting, reaching concentrations similar to rutin. As already observed for total phenolics, cooking resulted in higher amounts of the identified compounds.

### 3.4. E-Senses

E-sense data collected on cooked couscous samples was processed by PCA. The score plot and loading plot, in the plane defined by the first and second principal components (PC1 and PC2) that explained 84.17% of the variance, are shown in Figure 1.

Considering the score plot (Figure 1a), the couscous samples are well discriminated on PC1 and PC2 based on the sprouting time and enrichment level. The unsprouted sample (CTRL-C), composed of 100% pre-gelatinized buckwheat flour, is located on the negative part of PC1 (57.57% explained variance), well discriminated by all the other samples, and characterized by WC sensors (W1C and W3C) specific for aromatic and aliphatic compounds and by bitterness and astringency (taste and aftertaste) evaluated by e-tongue (Figure 1b). Couscous samples containing 25% and 50% sprouted buckwheat are mainly located in the positive part of PC1 and discriminated along PC2. In particular, couscous enriched with 25% sprouted buckwheat (S48h_25-C and S72h_25-C) are on the positive part of PC2 (26.6% explained variance), and both samples are characterized by WW e-nose sensors, specific for sulfur compounds, by the browning index (100-L*), and by umami and saltiness. Couscous samples with 50% sprouted buckwheat (S48h_50-C and S72h_50-C) are located in the negative part of PC2 and are mainly characterized by WS sensors of wide range sensitivity; in particular, the sample sprouted for 72 h (S72h_50-C) is perceived as the least bitter and astringent. Few studies are available on the evaluation of sprouting changes using e-senses. In the case of sprouted quinoa, e-tongue analysis pointed out an increase in sourness and a decrease in bitterness and astringency (both taste and aftertaste) upon sprouting for 48 h [6].

## 4. Discussion

The reformulation of conventional products with unconventional and sustainable raw materials is of great interest to the entire food system, from the valorization of resilient but underutilized crops to the increase in the availability of foods based on crops with unique compositional and nutritional characteristics. In this context, our work aims at the development and characterization of a 100% buckwheat product and at the understanding of the relationship between the compositional characteristics (specifically the phenolic compounds) and the sensory characteristics, evaluated by means of e-senses.

Buckwheat has been selected as a raw material thanks to its ability to be cultivated in a variety of environmental conditions, preferring humid and cool climates [37], and since it contains several bioactive compounds such as polyphenols and vitamins, whose benefits on human health have been recently reviewed [38]. Couscous was selected as the final product, appreciated all over the world, whose preparation requires short cooking times and a minimum amount of water [32], interesting aspects under the spectacle of limiting the environmental impact of food preparation.

Despite the benefits of sprouting on the nutritional value of grains [3,4], to date, there is little information about the impact of sprouting on product characteristics. Specifically, sprouted buckwheat has been added, as malt or sprouts, to muffins [30], cookies [39], and pasta [40]. However, to the best of our knowledge, no information is available on couscous or its sensory characteristics. Thus, to provide an overview of the impact of using sprouted buckwheat in the production of couscous, the reformulation of couscous included the addition of two levels (25% and 50%) of sprouted buckwheat at two sprouting times (48 h and 72 h). The physico-chemical properties of the products were evaluated, including cooking behavior and sensory features.

The physical characteristics that were mostly influenced by the presence of sprouted buckwheat were bulk density that generally increased, WAI that decreased, and WSI that increased (Table 2). The increase in bulk density suggests a greater aggregation of the particles; germination could have led to a change in the structural organization of the proteins, resulting in a greater ability to interact with each other. On the other hand, the changes in starch upon sprouting might account for the decrease in WAI and increase in WSI. Studies are underway to elucidate the changes in both starch and protein features in sprouted buckwheat and their relation to product quality.

The production process of couscous entails a first extrusion-cooking step and a subsequent stabilization of the final product [32]. The thermo-mechanical treatment determines the gelatinization of the starch, which organizes into an orderly structure during the cooling phase. Sprouting determines the hydrolysis of the starch granule and slows down both its gelatinization and the consequent retrogradation, as observed in several species, including wheat and quinoa [5,7]. The result is a lower structure of the product and a lower capacity to absorb water (lower WAI) and retain the material internally, with consequently higher solubilization (higher WSI). Such structure resulted, upon cooking, in a couscous characterized by low WAI (WAI before and after cooking are positively correlated; r = 0.97) and high WSI (WSI before and after cooking are positively correlated; r = 0.95), high hardness, gumminess, and resilience. Overall, except for the worsening in hardness, the addition of sprouted buckwheat improved the texture of the couscous in terms of cohesiveness, gumminess, and resilience, which increased with sprouted buckwheat. Changes occurring in starch structure upon sprouting (i.e., hydrolysis) would impact flour functionality, including its gelatinization and retrogradation properties, as reported for other grains, including pseudocereals [41]. The lower capacity of starch gelatinization and retrogradation in sprouted grains during couscous production might account for the differences in textural features between unsprouted and sprouted samples. Indeed, it is well known that in the case of gluten-free pasta-type products (e.g., couscous), the structure and texture of the product depend on starch gelatinization and retrogradation [42].

From a chemical standpoint, sprouting increased phenolics concentration in buckwheat seeds; from the literature, a significant increase in total phenolic content in the methanolic extract was observed after sprouting for 72 h (from 300 to 840 mg/100 g [15]) and 96 h (from 232 to 670 mg/100 g dw [33]). Consistent with literature data, the couscous formulations containing sprouted buckwheat showed higher TPC and higher antioxidant activity, with a positive relation to sprouting time and percentage of enrichment. The presence of sprouted flour also characterized the samples by higher rutin and quercetin concentrations, as well as by the presence of other flavonoids that could not be detected in the unsprouted sample (CTRL). Our data confirmed previous findings that showed an increase in flavonoids as the sprouting time increased [15,34,36].

Phenolic components and antioxidant activity increased upon cooking, which can be attributed to the higher extractability of the components from the matrix. Conversely, Carcea et al. [43] reported a 12% decrease in free total phenolics of buckwheat couscous after 6 min of cooking; however, the authors stated that, although statistically significant, the observed decrease in TPC might be due to the sensitivity of the method. Changes in TPC upon thermal treatments might be different depending on the type of treatment, e.g., amount of water and cooking time. For example, a decrease in free total phenolics up to about 20% was measured when cooking buckwheat flours in excess of water [14]. An overall decrease in TPC could be explained by either the solubilization of soluble components in the cooking water or their sensitivity to heat/oxygen/water treatment [43]. In the present study, the cooking process of the couscous consisted of a rapid (5 min) rehydration of the granules with hot water (at a maximum temperature of 100 °C) with no excess water; consistently, both dissolution and thermal damage phenomena are supposed to be limited.

To obtain a more comprehensive and exhaustive evaluation of cooked samples, their phenolic profile, antioxidant activity, and textural properties were combined with e-sensing data and processed by PCA in a global evaluation system that allows for the correlation of the smell, taste, and color of the samples with their compositional and textural characteristics. The PCA-biplot in the plane defined by PC1 and PC2 (80.07% explained variance) is shown in Figure 2.

The unsprouted sample (CTRL-C), composed of 100% pre-gelatinized buckwheat, is located in the negative part of PC1, completely separated from all the other samples mainly placed in the positive part of PC1, and discriminated on PC2 based on the enrichment level (25% and 50%). Considering the variable distribution on the plot, it can be observed that the unsprouted sample is described by e-nose and e-tongue variables. In particular, although characterized by the lowest content of total phenolics, CTRL-C was perceived as bitter and astringent.

The formulations with sprouted buckwheat, which are mainly described by the texture and chemical variables, by e-nose sensors (i.e., WW and WS), and by the browning index (100-L*), lay on the opposite side of the biplot. Among the enriched couscous, S72h_50-C is perceived as the least bitter and astringent, despite being characterized by the highest content of rutin and quercetin, typically described as bitter and astringent [10]. In the work of Borgonovi et al. [15], it was found that buckwheat sprouting increased the content of free phenolic compounds and antioxidant activity; on the contrary, it caused a decrease in total free flavan-3-ols, total bound phenolic acids, and total bound flavonols. This observation, combined with a different ratio of phenolic classes in sprouted and unsprouted samples, could explain the differences in the taste of couscous samples based on the percentage of sprouted buckwheat.

## 5. Conclusions

This work provides a multidisciplinary view of the effect of sprouted buckwheat enrichment on the chemical, physical, and sensory characteristics of couscous. The reformulation with 50% of buckwheat that was sprouted for 72 h resulted in a product richer in phenolic compounds (including rutin and quercetin), whose extractability was enhanced upon cooking, likely due to the modifications in interactions among various components. Moreover, in the presence of 72 h sprouted buckwheat, the couscous was characterized by improved texture characteristics (in terms of cohesiveness, gumminess, and resilience) and enhanced sensory attributes (with particular reference to reduced astringency, bitter taste, and aftertaste).

The relationship between phenolic compounds and sensory attributes also suggests that quercetin and rutin cannot be considered the only markers of bitterness/astringency in buckwheat matrices. Indeed, despite the increase in phenolic and flavonoid compounds in the products, bitterness and astringency decreased. Future work will aim at elucidating the impact of the sprouting process on the presence of peptides responsible for bitterness and astringency in buckwheat.

## Figures and Tables

**Figure 1 foods-12-03578-f001:**
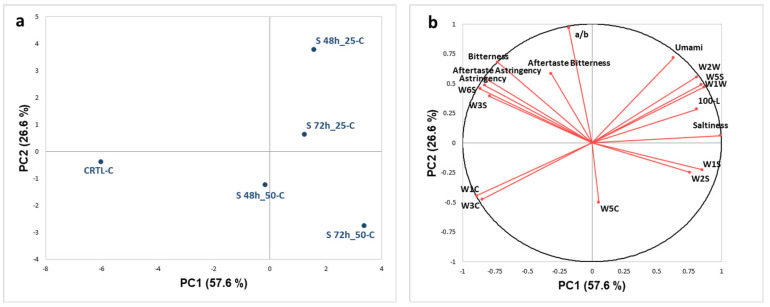
PCA score plot (**a**) and loading plot (**b**) of e-sense data collected on cooked couscous samples.

**Figure 2 foods-12-03578-f002:**
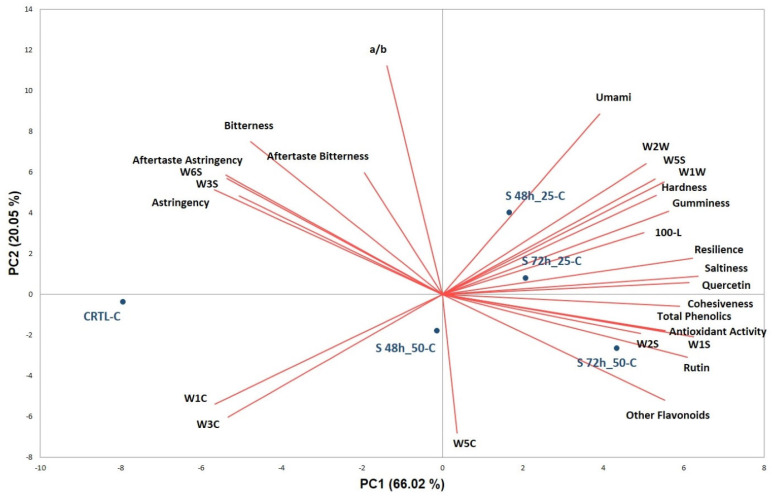
PCA-biplot of chemical, textural, and e-sense data collected on cooked couscous samples.

**Table 1 foods-12-03578-t001:** Couscous formulation and sample codes.

Couscous	Pregelatinized Unsprouted Buckwheat (%)	Sprouted Buckwheat for 48 h (S48h) (%)	Sprouted Buckwheat for 72 h (S72h) (%)
CTRL.	100	0	0
S48h_25	75	25	0
S48h_50	50	50	0
S72h_25	75	0	25
S72h_50	50	0	50

**Table 2 foods-12-03578-t002:** Bulk density, water absorption index (WAI), and water solubility index (WSI) of raw couscous.

Samples	Bulk Density (g/mL)	WAI (g/g)	WSI (g/g)
CRTL-R	0.647 ± 0.015 ^b^	4.07 ± 0.20 ^c^	4.45 ± 2.14 ^a^
S48h_25-R	0.695 ± 0.002 ^c^	3.60 ± 0.09 ^ab^	7.80 ± 0.58 ^b^
S48h_50-R	0.623 ± 0.005 ^a^	3.77 ± 0.12 ^b^	11.39 ± 0.55 ^c^
S72h_25-R	0.706 ± 0.002 ^c^	3.28 ± 0.03 ^a^	7.22 ± 0.13 ^ab^
S72h_50-R	0.659 ± 0.001 ^b^	3.77 ± 0.08 ^b^	9.86 ± 0.93 ^bc^

For couscous formulation, see Table 1. R = raw couscous. Mean (n = 3) ± standard deviation followed by different letters on the same column indicate significant differences (LSD test; *p* < 0.05).

**Table 3 foods-12-03578-t003:** Weight increase (WI), volume increase (VI), water absorption index (WAI), and water solubility index (WSI) of cooked couscous.

Samples	WI	VI	WAI	WSI
	(%)	(%)	(g/g)	(%)
CRTL-C	88.67 ± 0.92 ^n.s.^	27.42 ± 2.50 ^a^	2.15 ± 0.01 ^c^	1.98 ± 0.06 ^a^
S48h_25-C	90.43 ± 0.64 ^n.s.^	51.84 ± 5.24 ^b^	1.90 ± 0.04 ^ab^	4.16 ± 0.31 ^ab^
S48h_50-C	90.24 ± 0.43 ^n.s.^	42.03 ± 4.58 ^b^	1.95 ± 0.04 ^b^	6.93 ± 0.89 ^c^
S72h_25-C	89.05 ± 2.59 ^n.s.^	42.86 ± 0.01 ^b^	1.76 ± 0.06 ^a^	4.95 ± 0.70 ^bc^
S72h_50-C	90.16 ± 0.61 ^n.s.^	51.11 ± 3.85 ^b^	1.93 ± 0.10 ^b^	5.19 ± 1.73 ^bc^

For couscous formulation, see Table 1. C = cooked couscous. Mean (n = 3) ± standard deviation followed by different letters on the same column indicate significant differences (LSD test; *p* < 0.05); ^n.s.^, non-significant.

**Table 4 foods-12-03578-t004:** Texture profile analysis (TPA) of cooked couscous.

Samples	Hardness (N)	Cohesiveness	Gumminess (N)	Resilience
CTRL-C	6.24 ± 1.82 ^a^	0.46 ± 0.06 ^a^	3.06 ± 1.24 ^a^	0.27 ± 0.05 ^a^
S48h_25-C	14.97 ± 4.61 ^b^	0.57 ± 0.04 ^b^	8.53 ± 3.27 ^c^	0.38 ± 0.04 ^c^
S48h_50-C	7.86 ± 0.98 ^a^	0.56 ± 0.02 ^bc^	4.44 ± 0.65 ^b^	0.35 ± 0.01 ^b^
S72h_25-C	15.15 ± 2.76 ^b^	0.53 ± 0.03 ^b^	8.15 ± 1.83 ^c^	0.40 ± 0.03 ^c^
S72h_50-C	14.07 ± 2.81 ^b^	0.59 ± 0.03 ^c^	8.41 ± 2.06 ^c^	0.39 ± 0.02 ^c^

For couscous formulation, see Table 1. C = cooked couscous. The Mean (n = 21) ± standard deviation, followed by different letters on the same column, indicates significant differences (LSD test; *p* < 0.05).

**Table 5 foods-12-03578-t005:** Total phenolics, antioxidant activity, and individual flavonoids of raw and cooked couscous.

Samples	Total Phenolics (mg GAE/100 g dw)	Antioxidant Activity (µmol TE/100 g dw)	Rutin (mg/100 g dw)	Quercetin (mg/100 g dw)	Other Flavonoids (mg RE/100 g dw)
CRTL-R	62 ± 9 ^a^	148 ± 25 ^a^	2.02 ± 0.39 ^a^	n.d.	n.d.
S48h_25-R	75 ± 10 ^a^	220 ± 8 ^b^	3.63 ± 0.51 ^b^	0.39 ± 0.06 ^b^	1.33 ± 0.06 ^a^
S48h_50-R	141 ± 3 ^b^	373 ± 6 ^d^	6.04 ± 0.46 ^c^	0.60 ± 0.03 ^d^	4.05 ± 0.39 ^b^
S72h_25-R	132 ± 17 ^b^	444 ± 51 ^e^	7.15 ± 0.25 ^d^	0.54 ± 0.05 ^c^	7.02 ± 0.12 ^c^
S72h_50-R	256 ± 14 ^c^	730 ± 50 ^f^	12.88 ± 0.26 ^e^	1.20 ± 0.02 ^e^	9.45 ± 0.05 ^e^
CRTL-C	138 ± 6 ^b^	317 ± 24 ^c^	5.28 ± 0.28 ^c^	0.19 ± 0.01 ^a^	n.d.
S48h_25-C	331 ± 11 ^d^	871 ± 23 ^g^	15.51 ± 0.52 ^f^	1.23 ± 0.02 ^e^	8.03 ± 0.21 ^d^
S48h_50-C	414 ± 13 ^f^	951 ± 24 ^h^	17.09 ± 0.04 ^g^	1.19 ± 0.01 ^e^	13.63 ± 0.25 ^f^
S72h_25-C	354 ± 12 ^e^	976 ± 32 ^h^	21.65 ± 0.58 ^h^	1.54 ± 0.01 ^g^	19.64 ± 0.11 ^g^
S72h_50-C	373 ± 17 ^e^	1148 ± 30 ^i^	25.78 ± 1.20 ^i^	1.39 ± 0.02 ^f^	27.37 ± 0.12 ^h^

For couscous formulation, see Table 1. GAE = Galli Acid Equivalent; TE = Trolox Equivalent; RE = Rutin Equivalent; R = raw couscous; C = cooked couscous. n.d.: not detectable; Mean (n = 3) ± standard deviation followed by different letters in the same column indicate significant differences (LSD test; *p* < 0.05).

## Data Availability

The data that supports the findings of this study is included in the paper.

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
