# Peer review of "Reformulating Couscous with Sprouted Buckwheat: Physico-Chemical Properties and Sensory Characteristics Assessed by E-Senses"

_foods, 2023, doi:10.3390/foods12193578_

Round 1
Reviewer 1 Report
This study investigated the effect of two sprouting times (48 h and 72 h) and two enrichment levels (25% and 50%) of couscous prepared from buckwheat on physical properties, cooking behavior , chemical features and sensory traits.The authors found that the reformulation with 50% of buckwheat that was sprouted for 72 h resulted in a product richer in phenolic compounds, with improved texture characteristics and enhanced sensory attributes. Overall, the paper is well written and very comprehensive, with an extensive list of references.The study is interesting, however the paper needs some corrections/modifications.
1. There are many keywords that cannot reflect the main research content of the article well. It is recommended to reorganize.
2. Line 166-172,There is still a certain difference between an electronic eye and a color difference meter. Please clarify which instrument is used and choose the appropriate title.
3. Line 264,In terms of antioxidant capacity, only the total antioxidant capacity was measured, which is not sufficient to characterize the antioxidant strength of the substance. Other indicators should also be included, such as OH and DPPH free radical scavenging ability.
4. Line 306,It is recommended to analyze the original data of electronic nose, electronic tongue, and color difference before conducting principal component analysis to better explain the changes in relevant sensory and flavor characteristics.
5. In addition to the sprouting times and enrichment levels, the paper also compared related characteristics with the couscous and cooked couscous ,but the conclusion and abstract did not seem to reflect it. It is recommended to reorganize the abstract and conclusion.
Overall English writing of this paper is fine, but some details need to be improved, especially in the discussion of the results of the paper.
Reviewer 2 Report
Comments and Suggestions for Authors in the attachment.
